# Co-Electrospun Poly(ε-Caprolactone)/Zein Articular Cartilage Scaffolds

**DOI:** 10.3390/bioengineering10070771

**Published:** 2023-06-27

**Authors:** Andre M. Souza Plath, Stephanie Huber, Serena R. Alfarano, Daniel F. Abbott, Minghan Hu, Victor Mougel, Lucio Isa, Stephen J. Ferguson

**Affiliations:** 1Laboratory for Orthopaedic Technology, ETH Zurich, 8092 Zurich, Switzerland; huber.stephanie@hest.ethz.ch (S.H.); sferguson@ethz.ch (S.J.F.); 2Laboratory of Food and Soft Materials, ETH Zurich, 8092 Zurich, Switzerland; serenarosa.alfarano@hest.ethz.ch; 3Laboratory of Inorganic Chemistry, ETH Zurich, 8093 Zurich, Switzerland; dabbott@ethz.ch (D.F.A.); mougel@inorg.chem.ethz.ch (V.M.); 4Laboratory for Soft Materials and Interfaces, ETH Zurich, 8093 Zurich, Switzerland; minghan.hu@mat.ethz.ch (M.H.);

**Keywords:** PCL, zein, electrospinning, cartilage tissue engineering, chondrocytes

## Abstract

Osteoarthritis scaffold-based grafts fail because of poor integration with the surrounding soft tissue and inadequate tribological properties. To circumvent this, we propose electrospun poly(ε-caprolactone)/zein-based scaffolds owing to their biomimetic capabilities. The scaffold surfaces were characterized using Fourier-transform infrared spectroscopy, X-ray photoelectron spectroscopy, static water contact angles, and profilometry. Scaffold biocompatibility properties were assessed by measuring protein adsorption (Bicinchoninic Acid Assay), cell spreading (stained F-actin), and metabolic activity (PrestoBlue™ Cell Viability Reagent) of primary bovine chondrocytes. The data show that zein surface segregation in the membranes not only completely changed the hydrophobic behavior of the materials, but also increased the cell yield and metabolic activity on the scaffolds. The surface segregation is verified by the infrared peak at 1658 cm^−1^, along with the presence and increase in N1 content in the survey XPS. This observation could explain the decrease in the water contact angles from 125° to approximately 60° in zein-comprised materials and the decrease in the protein adsorption of both bovine serum albumin and synovial fluid by half. Surface nano roughness in the PCL/zein samples additionally benefited the radial spreading of bovine chondrocytes. This study showed that co-electrospun PCL/zein scaffolds have promising surface and biocompatibility properties for use in articular-tissue-engineering applications.

## 1. Introduction

Electrospun scaffolds have gained increasing attention as cell-secreted extracellular matrix (ECM) mimetic materials [1,2,3]. This makes them suitable for conjunctive tissue scaffolds such as cardiovascular [4], osseous [5], cartilaginous [6], and tendons [7]. Cartilage is a complex tissue in which the hierarchical structure, cell density, biochemical makeover, and shape determine the function [8,9]. Cartilage degeneration is a hallmark of osteoarthritis, a disease that affects the elderly worldwide [10]. To address this issue, poly(ε-caprolactone) (PCL) was electrospun with gelatin and sacrificial electrosprayed poly(ethylene glycol) [11] or chondroitin sulfate [12]. Matrix-assisted chondrocyte-grafted electrospun materials could fail due to low cell yield [6], delamination, and displacement [13]. These are associated with poor extracellular matrix mimicry and a lack of tribological properties. These materials have shown in vitro scaffolding properties. However, none of these studies investigated the effect of surface properties and protein adsorption, nor did they address the tribological properties. We hypothesized that the electrospun nanofibers based on PCL and zein can mimic the nanofibrous microenvironment of the cartilage extracellular matrix.

PCL is a semi-crystalline polymer approved by the Food and Drug Administration (FDA) for implantable contraceptive devices and resorbable sutures [14]. The lifespan of this polymer ranges from two to four years in vivo [15]. Chitosan [16], alginate [17], hyaluronic acid, collagen [18], silk fibroin [19], and chondroitin sulfate [12] have been mixed with synthetic polymers to decrease lifespan while increasing cell yield and chondrocyte gene expression in materials for articular cartilage tissue engineering [20]. By mixing these polyelectrolytes, it is possible to recreate the lubricating environment of cartilage by the entrapment of water molecules by surface charges [21,22,23]. We hypothesize that these charged microenvironments can recreate the bottlebrush-like surface features of articular cartilage promoted by the sulfated glycosaminoglycans of aggrecan and lubricin/PRG4. In this context, we propose blending zein with PCL, an amphiphilic corn protein rich in glutamic acid which is used in the pharmaceutic and food-packaging industries [24,25]. Zein solubility and other properties are directly related to the extraction method and this protein is considered safe for biomaterial applications by the United States Food and Drug Administration [25]. One-step electrospun zein blends have shown promising results with adipose-derived mesenchymal stem cells in a previous work [26]. Furthermore, zein was successfully incorporated with electrospun poly(l-lactic-co-glycolic acid)/hydroxyapatite and has shown human umbilical MSCs in vitro and in vivo cartilage regeneration potential [27].

We investigated the effects of hydrophilicity, surface chemistry, roughness, protein adsorption, and biocompatibility in electrospun scaffolds for articular cartilage tissue engineering seeded with primary bovine chondrocytes, and assessed their metabolism. We produced pure PCL and PCL/zein one-nozzle electrospun blends and characterized them using Fourier-transform infrared spectroscopy (FTIR) and X-ray photoelectron spectroscopy (XPS). We optimized the materials for their fiber diameter and increased zein content, as shown in the scanning electron microscopy (SEM) images. To date, this is the first report of up to 50% relative weight of zein in PCL electrospun matrices applied to articular cartilage tissue engineering.

## 2. Materials

Poly(caprolactone) (Mn 80 kDa) and analytical purity-grade solvents (formic and acetic acid) (FA/AA) were supplied by Sigma-Aldrich (Darmstadt, Germany). Zein (Mw 22 kDa) was supplied by Tokyo Chemical Industry (TCI, Eschborn, Germany). Cell culture media, Dulbecco’s modified Eagle medium (DMEM) F12, Fetal calf serum (FCS), antibiotics, and antimycotics (penicillin and streptomycin) were supplied by Gibco (Darmstadt, Germany). The cell-harvesting joint was supplied by a local abattoir. The enzymes for cell isolation, pronase, and collagenase P were supplied by Calbiochem (Darmstadt, Germany) and Roche (Basel, Switzerland), respectively. Horse synovial fluid obtained from the Tierspital (Zürich, Switzerland) was kindly obtained from the Center for Applied Biotechnology and Molecular Medicine (CABMM) at the University of Zurich. 

## 3. Methods

Electrospinning was performed using an EC-CLI device (IME, Waalre, The Netherlands). PCL was dissolved in the FA/AA binary solvent system and immediately electrospun to avoid cleavage of PCLs ester bonds. PCL and zein were dissolved in the same flask and spun immediately. We varied the weight per volume PCL (*w*/*v*) concentration and the relative zein mass concentration relative to the PCL content (wt.%). All PCL zein samples were electrospun at 18 kV by varying the distance from the collector to the spinneret (cm) and the flow rate (mL·h^−1^). For comparison, pure PCL samples were electrospun at 18 kV, 12 cm, and 1.0 mL·h^−1^ (Appendix A). All samples were spun at a −1 kV collector potential with Φ 60 × 180 mm^2^ and a 0.8 mm capillary for 3 h in a plastic syringe (Braun Inkjet, Melsungen, Germany) connected to a poly(tetrafluoroethylene) with a Φ 1.5 mm tube.

We assessed fiber diameter using an electron microscope (SEM, Hitachi SU5000, Japan). Samples were cut with scissors, glued to the stubs with carbon tape, and sputter-coated with 4 nm platinum–palladium (Pt-Pd 80%/20%, CCU 010 Safematic, Switzerland). Field emission scanning electron micrographs (FE-SEM or SEM) at 5000× magnification was performed at low voltages (<5 kV) and spot intensities to preserve the sample topography and reduce charging effects on the polymer surface. Images were obtained with a secondary electron detector in high-vacuum mode with a distance to the probe inferior to 10 mm. Finally, we measured the fibers using the ImageJ software. Briefly, 50 random nanofibers were measured from 5000× magnification images. The descriptive statistics and comparison between fiber diameters were made according to the protocol described below. 

The surface chemistry was evaluated using Fourier-transform infrared spectroscopy in total attenuated reflectance (FTIR-ATR) mode and X-ray photoelectron spectroscopy (XPS) (Thermo Fischer, Alpha 110, Karlsruhe, Germany). FTIR spectra were acquired using Varian 640 (Agilent) apparatus at room temperature, as the average of 32 scans between 4000 and 600 cm^−1^ with 4 cm^−1^ resolution. XPS was carried out using a monochromatic aluminum source, spherical analyzer, and multichannel detector. The high-resolution spectra were recorded using a 135 W source power, 23.5 eV analyzer pass energy with 0.1 eV steps, the X-ray spot at 800 µm of the sample, and the photoelectron takeoff angle of 45° For the survey and HR spectra, we acquired 3 and 81 signal accumulations, respectively. The data were analyzed using Casa XPS software. The spectra were normalized to the C1s adventitious carbon at 285.0 eV. Peak deconvolutions were performed to best fit the experimental data using the Shirley background. 

The samples were washed, peeled off the aluminum foil, and mounted onto glass slides with carbon tape. The static water contact angles (WCAs) were measured using a sessile goniometer (Kruss DSA100, Hamburg, Germany) at 0 s. The droplet shape was captured by a camera and analyzed using the DSA4 software (Kruss, Germany). The experiments were performed in triplicates for each of the substrates in different areas. 

The samples were washed, peeled off the aluminum foil, and dried in a glass Petri dish. They were then analyzed using an optical 3D profilometer (PluNeox, Sensofar, Spain). Individual areas of the sample were focused on using a 20× magnification lens (NA 0.45) and taken with the manufacturers’ software. They were analyzed using Gwyddion 2.0 open-source software without leveling, from uniform areas of the sample. The average roughness parameters (R_a_) were obtained from a 200 × 250 pixel (170 × 210 μm). 

The samples (PCL, 8P40Z, or 6P50Z) were washed, peeled off the aluminum foil, and dried in a glass Petri dish. Circular specimens with a diameter of 10 mm were punched out and washed in phosphate-buffered saline (PBS) at room temperature for 24 h. The protein solution was freshly prepared by either dissolving bovine serum albumin (BSA) (Panreac Aplichem, Darmstadt, Germany) in PBS to obtain a final concentration of 500 µg/mL or by diluting horse synovial fluid 50-fold in PBS. Each specimen was incubated with 130 µL of protein solution at 37 °C for 24 h. For the reference measurement, an additional protein solution was incubated without contact with any specimen. After incubation, the specimens were removed from the solution and the protein content was measured using the indirect bicinchoninic acid (BCA) assay (Pierce™ BCA Protein Assay Kit, 23225, Thermo Scientific, Switzerland) following an enhanced measurement protocol. Briefly, 25 µL of diluted sample or standard was combined with 200 µL of BCA reagent work solution and incubated at 60 °C for 30 min. The absorbance of the solution was measured at 562 nm using an Infinite M200 PRO plate reader (Tecan, Switzerland). Measurements were performed in triplicate. The protein content was quantified based on the protein assay standard curve and subsequently, the protein adsorption for each specimen (sample size = 5) was derived.

Bovine chondrocytes were isolated from three different donors from an ox knee tibia and femur, as previously reported by Walser et al. [28]. The joints were washed and immersed in a betadine and water solution. They were dissected to expose the articulating surface, and small pieces of tibia and femur surface cartilage were cut with a scalpel. The dissected pieces were immersed in a 3% vol. PBS antibiotic/antimycotic solution. They were washed, minced, and digested with 0.4% vol. pronase (Calbiochem) for 90 min, and posteriorly with 0.025% vol. collagenase P (Roche) overnight under incubation (37 °C, 5% CO_2_) and frequent gentle stirring. Nucleated cells were harvested using a 100 μm cell strainer, centrifuged, counted, and frozen in 10% DMSO, 40% FCS, and DMEM F12 freezing media until use in liquid nitrogen. 

Cell cultures were performed in 6 mm diameter biopsy punched pre-conditioned membranes. We washed the membranes with water and sterilized them for 2 h under UV light in the biological safety cabinet. They were then immersed overnight in a PBS solution containing 3% antibiotics/antimycotics. Finally, the membranes were pre-conditioned for 1 h in cell culture media (DMEM F12 supplemented with 10% Fetal Calf Serum (FCS) and 1% antibiotics/antimycotics. The harvested cells were thawed and placed in a T75 flask for one week, trypsinized, and resuspended for cell seeding. Bovine chondrocytes (passage 1) were seeded onto the scaffolds at a 5.000 cells/cm^2^ density for 15 μL per scaffold. Cell metabolic activity was determined using the PrestoBlue Assay (Thermo Fisher, Germany) with fluorescence measurements at 584 nm in a multi-plate reader. Cells were cultured for 7 days and 14 days, fixed in 4% paraformaldehyde for 20 min, and stored in PBS microscopy at 7 °C. For staining, permeation was performed with 0.5% Triton-X for 15 min. Non-specific dye interactions were blocked with 1% bovine serum albumin protein (Panreac Aplichem, Germany) at room temperature for 2 h. Samples were then stained with 4,6-diamidino-2-phenylindole (DAPI) (Thermo Fisher, Germany) (nuclei) and Alexa Fluor 568 Phalloidin (Thermo Fisher, Germany) (F-actin), washed with PBS and transferred to a new 48-well plate. Images were obtained at a 20× magnification (NA 0.4) using an inverted phase contrast microscope Olympus IX51 coupled with a camera (Appendix A). Widefield Fluorescence Microscopy Image acquisition and processing were made with the manufacturer’s software (CellSens, Olympus) with exposure time under 200 ms, and gain under 2× with the reflected light (fine) mode. Finally, confocal laser microscopy measurements were performed using a Leica LSM780 upright system (Wetzlar, Germany). Briefly, the samples were mounted onto a glass slide with VectaShield (Vector Laboratories, Burlingame, United States). A coverslip was then placed. Slides were sealed using nail polish. The samples were mounted on the microscope and acquired with 20× magnification, 405 and 568 nm, at a 2148 × 2184 pixels density and 6.0 μs pixel dwell with the manufacturer’s software (Zeiss Zen).

Fiber diameters (*n* = 50 measurements), roughness (*n* = 3 membranes), and WCAs (*n* = 3 membranes) were analyzed for the mean and standard deviation using Prism GraphPad version 9.0 (GraphPad Software Inc., La Jolla, CA, USA). One-way ANOVA and Tukey’s post hoc tests were performed to evaluate statistical significance. The distribution of protein adsorption data (*n* = 5 samples per group) and the metabolic activity (*n* = 3 samples) were assessed with a QQ-plot and subsequently analyzed using Brown-Forsythe and Welch ANOVA test followed by Dunnett T3 multiple comparisons post hoc tests with the same software.

## 4. Results

### 4.1. Nanofiber Mats Characterization

Nanofibers were optimized for their morphology by varying the flow rate (0.5 and 1.0 mL·h^−1^), distance to the collector (12 and 16 cm), and zein content (40 wt.% and 50 wt.% relative to PCL) according to the experimental design in Table 1. In the given parameter space, all the conditions were spinnable for at least one hour without reproducibility issues. The samples were coded with letters A to H. The surface morphology can be seen in Figure 1.

The PCL/zein blends show nanofiber diameters in the range between 146 nm and 266 nm. To establish a comparison, a pure PCL sample was electrospun with the given conditions in Section 3. The reference PCL sample showed an average diameter of 156 ± 13 nm. We observed that an increase in flow rate resulted in lower fiber diameters under all given conditions. This effect may be attributed to better jet elongation and solvent evaporation. The same cannot be drawn for an increase in the distance to the collector—it was not possible to draw inferences from the experimental conditions in Table 1. For the first time, nanofibers with 50 wt.% zein were spinnable in a one-step process. Compared with the literature [26], the nanofibers containing 40 wt.% PCL were at least 80 nm finer. This might be related to the fact that the process voltage was increased from 12 to 18 kV and that the flow rate was increased from 0.5 to 1.0 mL·h^−1^. For spectroscopy and cell studies and to reduce the influence of nanofiber diameter, the finest PCL/zein samples were chosen—sample B (146 ± 11 nm) and sample H (168 ± 12 nm). 

The XPS survey spectra show a 2.7 (Table 2) carbon-to-oxygen ratio in the electrospun PCL samples, as previously observed in the literature [26,29] (Appendix A). The survey spectra revealed the presence of zein in the electrospun mats, as evidenced by the appearance of the N1s signal at ca. 400.0 eV. An increase in the zein content promoted an increase in the N1s signals from 9.8% in sample 8P40Z to 11.6% in sample 6P50Z (Table 2). The nitrogen content is slightly higher than previously reported (6.4%) [26] because of the electrospinning for up to three hours. This promoted finer fibers, and thus higher surface area and an abundance of charged chemical groups. High-resolution C1s spectra show the expected ester, carbonyl, and aliphatic carbon peaks at 288.8, 286.5, and 284.9 eV, respectively, for pure PCL, as shown previously by Louette et al. [29]. For O1s, ester and carbonyl peaks were observed at 532.7 and 531.3 eV (Figure 2). These values are 0.8 eV lower than previously reported by Louette [29] films cast on CHCl_3_. This might be attributed to the charging correction of C 1s spectra to 285.0 eV. For the PCL/zein electrospun blends, ester bonds are combined with the amides from the primary structure of the protein as seen in the amide peaks at 287.5 eV (Figure 2). N 1s high-resolution spectra (Appendix A) show that the surface might have protonated amines and amides with the same percentage area (50.0%, Appendix A) and binding energies at 399.7 and 399.4 eV, respectively (Figure 2). An increase in protein content resulted in the doubling of the amide peak area both in C 1s and O 1s spectra (Table 3). The small shifts for the assigned peaks are negligible (<0.5 eV) (Figure 2).

FTIR-ATR was performed on the electrospun membranes to further investigate the conformational effects of zein on the blends. We observed an increase in the absorbance of the broad carboxyl and amide bands at 3305 cm^−1^ (Figure 3). As expected, Amide I and Amide II bands at 1655 cm^−1^ and 1541 cm^−1^ originated from zein. Interestingly, the contribution of amide I is decreased with increasing the protein content. Finally, we observed a decrease in the absorption in the regions corresponding to the –COC and –CO regions (1722 and 1166 cm^−1^, respectively). This evidence indicates that the reorganization of zein in polyester/protein blends expose primary amino groups from the primary structure of the protein. These findings were previously reported by the authors. We hypothesized that, in this case, the increase in primary amino groups, possibly protonated, correlates with the behavior change compared with pure PCL. Figure 4 shows the static water contact angle (WCAs) (t = 0 s) reduced from 118° ± 7° to 61° ± 4° and 56° ± 2°. 

Figure 5 shows the 3D profiles of the PCL, 8P40Z, and 6P50Z samples with Ra values of 678 ± 165, 459 ± 91, and 477 ± 160 nm, respectively. Despite the similar values, there was a statistically significant difference (*p* < 0.05) in the inclusion of zein in the blends. We hypothesized that the amphiphilic protein associated with the formic/acetic acid binary solvent system promoted a higher elongation of the polymer jets, promoting a uniformly smoother surface.

### 4.2. Protein Adsorption and Cell Cultures

Figure 6 presents the protein adsorption indirectly measured for PCL, 8P40Z, and 6P50Z scaffold specimens (10 mm diameter). Protein adsorption on pure PCL scaffolds is significantly higher than that in 8P40Z (*p* = 0.021) and 6P50Z (*p* < 0.001). Average BSA adsorption was 31.24 ± 7.24 µg on the PCL scaffold, 10.67 ± 11.00 µg on the 8P40Z specimen, and 2.73 ± 4.49 µg on the 6P50Z specimen. Although the average protein adsorption was reduced in 6P50Z compared with 8P50Z, the difference is not statistically significant (*p*-value = 0.409). For synovial fluid, average values were 21.39 ± 2.15 µg, 8.82 ± 3.00 µg, and 10.19 ± 3.73 µg for PCL, 8P40Z, and 6P50Z, respectively. The results follow the same trend as for BSA; however, there is a slight increase in average protein adsorption in the sample comprising more zein.

Figure 7 shows the growth and spread of stained P1 bovine chondrocytes onto the electrospun scaffolds at D7 and D14. We noticed that more cells were present in zein-comprised samples (8P40Z and 6P50Z) on day 14. For F-actin staining with Alexa Fluor 568, we noticed that the cells spread more radially when zein comprised the materials, whereas F-actin filaments possibly aligned to nanofibers in the PCL samples. Cell viability (Presto Blue Assay^®^, Thermo Fischer, Germany) increased up to day 14 in all scaffolds. It is worth mentioning that zein-comprised samples had increased average metabolic activity compared with pure PCL at all time points (Appendix A). F-actin staining on Day 14 indicates that the initially seeded cells proliferated until they covered the entire surface of the materials. 

## 5. Discussion

For the first time, PCL/zein co-electrospun samples with diameters under 200 nm have been produced. Compared with previous studies [26], this has been achieved by reducing the amount of PCL in the blends and increasing the amount of zein. In a similar study, Plath et al. [26] uniformly spun 10% PCL with 30 wt.% zein (363 ± 44 nm) and 8% PCL with 40 wt.% zein (220 ± 40 nm). The smaller fiber diameters (under 200 nm) obtained in this study are related to the increased voltage in the positive pole (18 kV) compared to the previous study (12 kV) and climate control of the chamber. These parameters synergistically result in higher fiber drawing, jet splitting, and appropriate solvent evaporation [30]. The results affected the lower surface roughness, which is still a poorly explored parameter in the electrospinning field. In scaffolds for articular cartilage tissue engineering, diameter reduction to dimensions closer to the underlying collagen II nanofibers is desired and is a promoter of the chondrocyte phenotype [19,31]. However, Jeong and Hollister [32] pointed out that material choice and architecture are critical aspects of cartilage scaffolds. To date, most studies have investigated the effect of fiber diameter, but little has been reported on the reproducibility of the samples and surface roughness effects. In this study, the surface roughness was assessed using a profilometer, which is limited by the Rayleigh criterion to the numerical aperture of the lens and the wavelength [33]; therefore, individual nanofibers cannot be resolved. Roughness assessments were also performed in other works with comparable diameters to Figure 1, using atomic force microscopy [34]. However, this technique is limited to a smaller scanned area of the sample. Dubey et al. [34] could resolve single fibers; however, in their work they could only analyze a 10 × 10 μm area. Similarly, Mozzafari et al. [35] analyzed the roughness of a 5 × 5 μm modified-gelatin nanofibers and could resolve single fibers. It is well-known that surface roughness can benefit the retention of synovial fluid and the promotion of lubrication in articular cartilage [36,37]; however, macrorough surfaces can hinder contact guidance and thus cell adhesion and spreading [38]. The measured values are comparable to the interferometry results (approximately 500 nm) reported by Espinosa and collaborators in the bovine tissue [39]. 

Based on the FTIR-ATR and XPS data, we hypothesized that the heavily charged moieties of the glutamic acid-rich protein were drawn by the positively charged terminal in which the spinneret was connected, segregated to the surface, and drawn by the electric field. It is well reported that the lower molecular weight component of an immiscible polymer blend segregates to minimize surface energy of the system [40]. Similar surface segregation phenomena have been studied by Kurusu with Pluronic (PEO-PPO-PEO) block copolymers blended with PVDF [41] and SEBS [42]. The authors investigate the surface organization of the hydrophilic PEO moieties on the surface of the electrospun scaffolds. To verify this hypothesis, zein was incubated at room temperature overnight in ethanol. After at least 12 h, the FTIR data (Appendix A) showed a reduction in the amide I peak (1722 cm^−1^) and complete removal of the amide I peak (1638 cm^−1^),indicating not only zein concentration on the surface of the electrospun mats, but also that the immiscible protein was weakly bound to PCL in the matrices. Similarly, the amide peak areas for the XPS data shown in Table 2 confirms the hypothesis of zein surface enrichment by surface segregation. This might be due to the electrospinning process in FA/AA. This would then impart charged chemical microenvironments to attract and retain water and change the hydrophobic behavior of the PCLs. Similar results are pointed out in co-electrospun synthetic/biopolymer blends with silk-fibroin ([43]) and chitosan ([44]), for example.

Interestingly, the changes in surface chemistry affected the protein adsorption in the tested groups. Scaffolds with zein showed lower protein adsorption when exposed to bovine serum albumin and equine synovial fluid (Figure 6). Protein adsorption is a complex dynamic phenomenon governed by secondary interactions (van der Waals, hydrophobic, hydrophilic, and electrostatic interactions) and protein folding [45,46]. The adsorption/desorption of multi-protein solutions usually consists of a reversible phase for proteins with higher mobility (lower Mw) and an irreversible phase for lower mobility proteins (higher Mw), which unfolds during the process [46,47]. We hypothesized that the charged groups of surface-segregated zein and the hydrophilicity of the membranes favored water adsorption to the detriment of proteins. Similar behavior was observed in Li et al. [48]. Here, the authors electrospun PLGA and grafted chitosan to the surface using alkaline hydrolysis via NHS: EDC chemistry. The immobilization of chitosan benefited not only in hydrophilicity but also implied a decrease in BSA proportional to the amount of chitosan [48]. Similar results are also reported for synthetic/synthetic bicomponent blends, e.g., PLGA and star-shaped poly(ethylene oxide) [49]. In the context of biomaterials, hydrophilicity, roughness, and protein adsorption mediate cell-material interactions [47,50]. The optimization of these parameters is specifically desired in tissue-engineering applications, but on the other hand, can evoke a negative host response and reduce the biocompatibility of a material surface [51]. Here, we found reduced protein adsorption, which however did not negatively affect cell-scaffold interactions, metabolic activity, and proliferation in vitro. 

Ethanol and heat sterilization were not possible since zein would be solubilized [24] (Appendix A) or the blend would melt (T_m_ 55 °C) [52]. To eliminate artifacts of cell culture on imaging (e.g., detaching, and non-adherent cells growing in the wells), the scaffolds were transferred to a new 48-well plate for staining and later mounted onto slides. We hypothesized that cell spreading is favored by the smoother surface as well as the change in surface free energy (i.e., hydrophilicity). Additionally, the higher hydrophilicity of the zein-containing samples (WCAs of approximately 60° compared with 120° of pure PCL) can mimic the native ECM makeover, composed of aggrecan and lubricin, providing appropriate cues for cell spreading and proliferation. The presence of hydrophilic groups is beneficial in the review conducted by Amani [51]. The adsorption of proteins, which is considerably higher in PCL, may also create a synergistic effect with the previously mentioned variables. The cells in the zein-comprised membranes presented rounded morphology and multiaxial spreading to the detriment of the monoaxial alignment in the samples with PCL, which might be attributed to a synergistic effect of the surface chemistry, protein adsorption, and reduced roughness. This might be an indication of phenotype maintenance, as observed in [53]. RT-PCR needs to be conducted in a follow-up study. With this technique, the expression of chondrocyte phenotype markers such as collagens I and II and aggrecan performed by Li et al. [53] could be appropriately assessed. Widefield Fluorescence Optical microscopy has as the main disadvantage an uncontrolled fluorophore excitation (increased background) and the inability to resolve multiple layers of the three-dimensional scaffold structure simultaneously without imaging processing [54] (Appendix A). To circumvent these issues, imaging was performed at the lowest exposure and with a lower gain to reduce imaging artifacts. The morphology of the cells and nuclei were observed under the given conditions. 

Overall, the study findings indicate promising biocompatibility and surface properties of PCL/zein scaffolds compared with pure PCL scaffolds. However, it must be noted that in vitro conditions may differ from in vivo conditions, and, furthermore, the adsorption of some specific proteins could be beneficial. For instance, adsorption of the lubricating protein (proteoglycan 4), which is produced by chondrocytes but is also found in synovial fluid [55], should improve frictional properties of the scaffolds and thereby promote articulation with tissue-engineered cartilage grafts. Therefore, further studies are required to comprehensively assess the biomechanical performance of PCL/zein in cartilage tissue engineering. 

## 6. Conclusions

Zein produces beneficial effects on scaffolding materials for articular cartilage tissue engineering. For the first time, nanofibers smaller than 200 nm were produced using a one-nozzle process. Surface chemistry assessment using spectroscopy has shown that zein reorganizes to the surface of electrospun materials, imparting low protein fouling and hydrophilicity. The materials composed of zein exhibited lower surface roughness. Based on this, we hypothesized the synergistic effects of protein adsorption, roughness, and hydrophilicity on cell morphology and metabolism after 7 and 14 days of culture. 

## Figures and Tables

**Figure 1 bioengineering-10-00771-f001:**
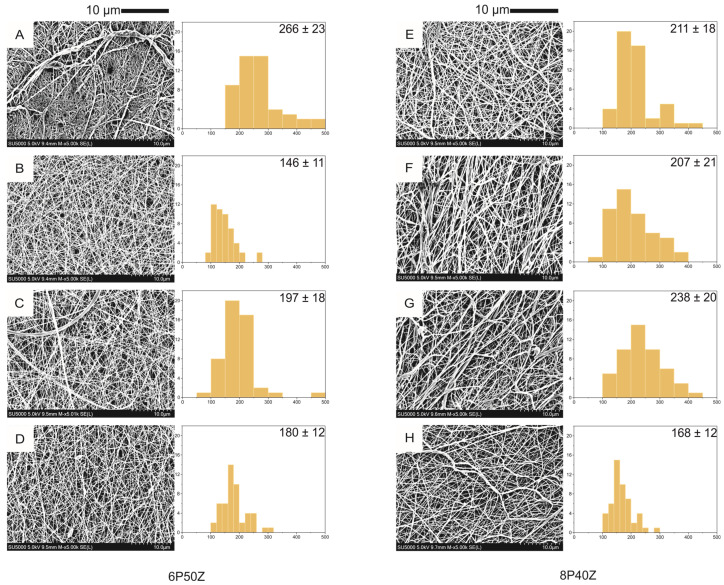
Nanofiber morphology of PCL/zein electrospun mats prepared from a 70/30 vol./vol. FA/AA conditions from Table 1. Average fiber diameter (*n* = 50 measurements) reported in nanometers.

**Figure 2 bioengineering-10-00771-f002:**
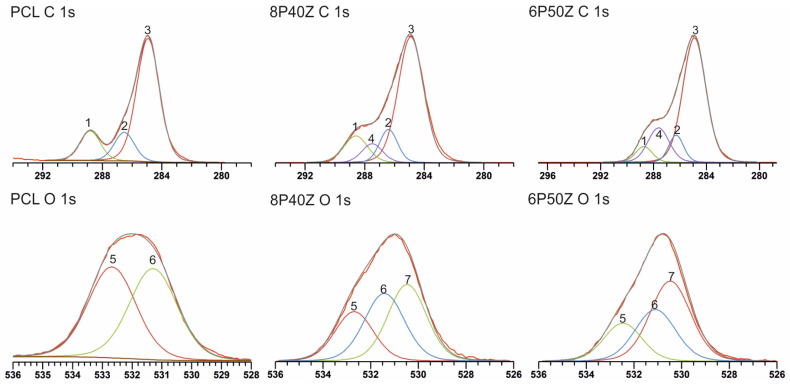
C1s and O1s X-ray photoelectron spectra of the electrospun PCL, 8P40Z (H), and 6P50Z (B) samples. Peak label information in Table 2.

**Figure 3 bioengineering-10-00771-f003:**
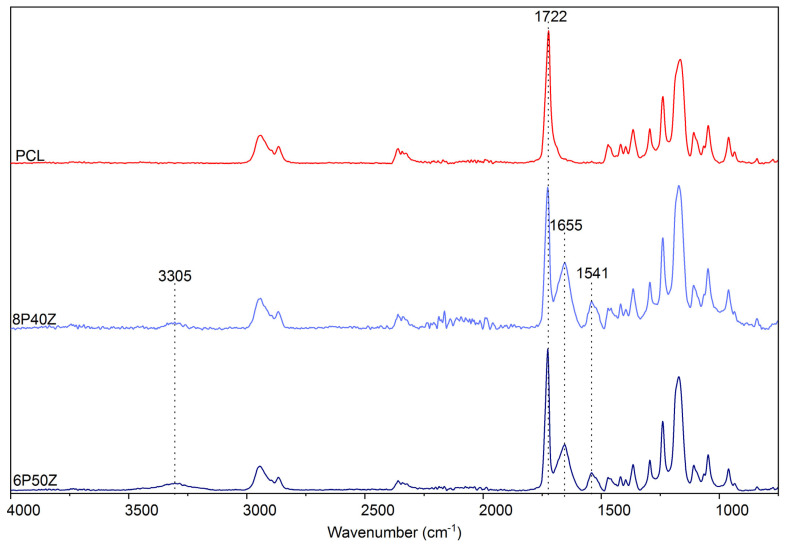
FTIR−ATR performed on the electrospun samples (PCL, 8P40Z, 6P50Z) prepared from a 70/30 *v/v* FA/AA solution.

**Figure 4 bioengineering-10-00771-f004:**
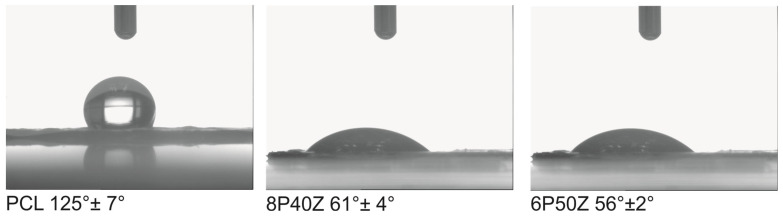
Static Water Contact Angles (t = 0 s) performed on the PCL/zein electrospun surfaces.

**Figure 5 bioengineering-10-00771-f005:**
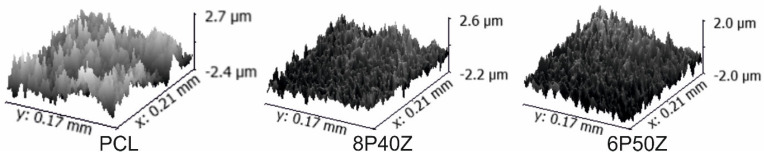
Images obtained with the PluNeox Profilometer at 20× magnification lens analyzed using Gwyddion 2.0.

**Figure 6 bioengineering-10-00771-f006:**
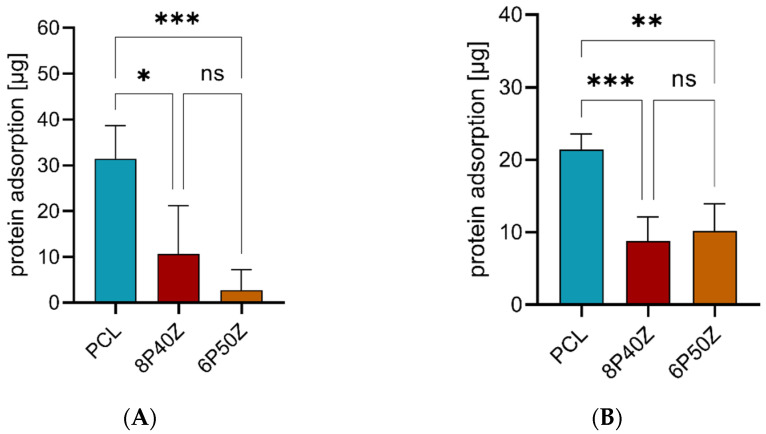
Protein adsorption on PCL, 8P40Z, or 6P50Z scaffold specimen measured using indirect BCA assay. (**A**) Adsorption of BSA and (**B**) adsorption of synovial fluid proteins to scaffold specimen (sample size 5). Statistically significant differences between groups are indicated by * where *p* ≤ 0.5, ** where *p* ≤ 0.01, and *** where *p* ≤ 0.001.

**Figure 7 bioengineering-10-00771-f007:**
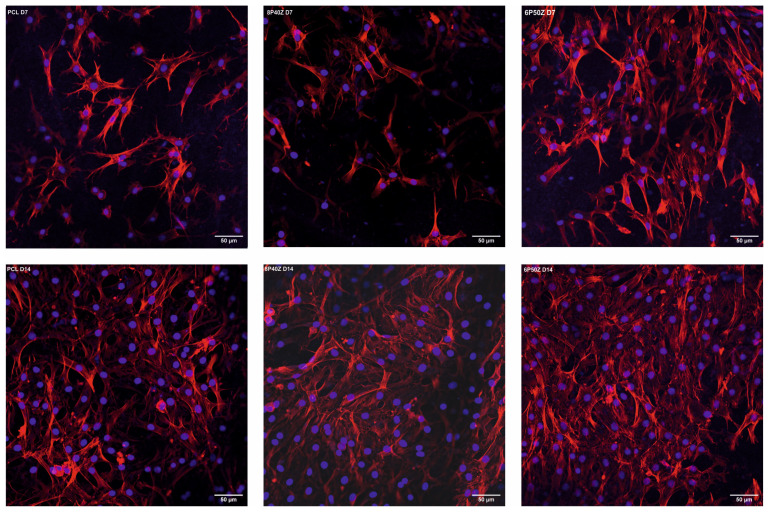
20× magnification laser scanning confocal microscopy of PCL, 8P40Z, and 6P50Z seeded with P1 bovine chondrocytes at days 7 and 14, stained with DAPI (blue) and Alexa Fluor 568 Phalloidin (red). Scale bars: 50 µm.

**Table 1 bioengineering-10-00771-t001:** PCL/zein electrospinning conditions of the membranes in Figure 1.

	Sample	Flow Rate (mL·h^−1^)	Distance (cm)	PCL(g)	Zein (g)
06P50Z	A	0.5	12	0.240	0.120
	B	1	12	0.240	0.120
	C	0.5	16	0.240	0.120
	D	1	16	0.240	0.120
08P40Z	E	0.5	12	0.320	0.128
	F	1	12	0.320	0.128
	G	0.5	16	0.320	0.128
	H	1	16	0.320	0.128

**Table 2 bioengineering-10-00771-t002:** Elemental composition of PCL, 8P40Z, and 6P50Z nanofiber mats. C: carbon, O: oxygen, and N: nitrogen.

	Atomic Concentration of Surface Detected Elements (%)				
Sample	C1s	O1s	N1s	C/N	C/O
PCL	73.3	26.7	0	X	2.7
8P40Z	67.3	23	9.7	6.9	2.7
6P50Z	66.6	21.8	11.6	5.7	2.9

**Table 3 bioengineering-10-00771-t003:** C1s and O1s High-resolution spectra data of the electrospun PCL, 8P40Z (H), and 6P50Z (B).

High-Resolution Spectra Information						
C 1s Spectra data						
Peak	PCL BE (ev)	PCL Peak Area	8P40Z BE (eV)	8P40Z Peak Area (%)	6P50Z BE (eV)	6P50Z Peak Area (%)
1 (O=C–O)	288.8	15.2	288.7	13.2	288.7	6.2
2 (C–O)	286.5	14.4	286.4	12.4	286.6	9.5
3 (C–H/C–C)	284.9	70.4	284.9	66.2	284.9	67.1
4 (O=C–N)			287.5	8.3	287.6	17.1
O 1s Spectra data						
Peak	PCL BE (ev)	PCL Peak Area	8P40Z BE (eV)	8P40Z Peak Area (%)	6P50Z BE (eV)	6P50Z Peak Area (%)
5 (O=C–O *)	532.7	50.0	532.7	25.1	532.5	21.2
6 (* O=C–O)	531.3	50.0	531.4	35.9	531.1	30.7
7 (* O=C–N)			530.5	39.0	530.5	47.9

* BE: Binding Energy.

## Data Availability

Data supporting this study are included in within the article and/or Appendix A.

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
