# Peer review of "Co-Electrospun Poly(ε-Caprolactone)/Zein Articular Cartilage Scaffolds"

_bioengineering, 2023, doi:10.3390/bioengineering10070771_

Round 1

Reviewer 1 Report

bioengineering-2459878

Title: Nanofiber surface of poly(ε-caprolactone)-zein-based one-nozzle electrospun articular cartilage scaffolds

Overview

The work addresses the production of PCL fibers enriched with zein protein in order to form a mesh for cartilaginous tissue engineering. In general, the work is well written and the data well presented.

Next, I will make some suggestions for enriching the work. The evaluation was carried out through the eyes of a researcher who is familiar with the characterization of biomaterials, but whose specialty is in vitro and in vivo assays.

Introduction

The Introduction is well-written and well-contextualized. The issue of tissue engineering applied to cartilaginous tissue is, in my view, well placed. The PCL is well covered. One suggestion would be to address the zein protein a little more. I leave that to the discretion of the authors.

Materials and methods

A comment to the authors.  Electrospinning is a widely used and reproducible method for fiber formation. But, in general, it is slow and with not very adequate performance. Thinking about clinical use, this may be a limiting factor. As the data presented in this paper point to an interesting product, it is worth the authors to find out (on another occasion, of course) about other techniques for fiber production, such as rotary jet spinning or solution blow spinning.

Regarding the production of fibrous scaffolds (between lines 80-86), the PCL and zein masses used are in table 1. But maybe they should be in M&M. Perhaps the different samples produced (PCL, 8P40Z, or 6P50Z) should be at this point as well, to guide the reading.

For tissue engineering, sterilization is always an issue. I agree with the form chosen in this study (line 149). However, do the authors have any information about the quality of samples after sterilization? Well, 2h in UV is a significant amount of time. Also, was the UV used in the biological safety cabinet? Normally, this equipment uses UV-C, less energetic. Other lamps have higher energy (UV-A and UV-B). Sterilization can be more efficient, but it can bring changes to the samples (from degradation to cross-linking). Perhaps the authors, on another occasion, can analyze this issue. And perhaps, look into other forms of sterilization (gamma radiation could be an alternative).

About the use of cells (bovine chondrocytes) I have a doubt. Between lines 145-147 it was said that the cells were frozen after their extraction and isolation. But between lines 153 and 157 it says that fresh chondrocytes are used. I wonder what the correct way is.

In the same context, line 154 reads “Bovine chondrocytes (passage 0)”. If the cells were indeed frozen, considering that chondroblasts are anchorage-dependent cells, they were released from the growth substrate for freezing. So, technically, a pass was made. I don't think this interferes with the results. But I understand that authors should register. I suggest “pass 1” or “pass 0, post cell thawing” for clarity, if applicable.

Alexa Fluor 568 (lines 62-63) is a broad class of components, including fluorescent dyes and antibodies. I suggest using the Alexa Fluor 568 Phalloidin nomenclature (here and throughout the rest of the text) for greater accuracy. The results show Alexa Fluor 568, but the caption for figure 6 says Phalloidin only.

Results

The way to measure fiber diameter was not clearly described in the M&M. Perhaps this can be revised. As for the results (lines 196-208 and figure 1), I understand from the figures that the diameter was established and presented in 50um intervals (eg 100um, 150um, 200um, etc.). If so, perhaps the average is not the best way to present the data. Perhaps the authors should work with the mode value of the most common range. Here is the suggestion for reflection by the authors. If the authors made the measurement at intervals smaller than 50um, I think the graphs could be kept as they are, but the intermediate values could be presented in a new table.

I confess that I found it strange to combine the FTIR and the contact angle in the same figure (Figure 3). I don't think the text has been corrected. I suggest separating and, if applicable, perhaps rearranging the contact angle figure so that the figure is horizontal. I think it looks more elegant and would take up less space.

The protein adsorption data (lines 63-69, figure 5) are very interesting. I agree with the authors that this does not seem to have implications after 24 hours, as data on morphology and viability go in that direction. But do the authors have any information about initial adhesion? With just a few hours of incubation? It's theory, adherence could be slower. It is suggested for future assays.

Between lines 279-280 the cell viability results are commented. But there is little information, and this is important data. In fact, as this is an assay with 1, 7 and 14 days, in addition to viability, it brings information about cell growth as well. I suggest moving this data from the supplementary material into the main article. By the way, it is worth indicating in the figure the groups that are significantly different.

Discussion

The Discussion is written and contextualized. I have no comments to make.

Conclusion

The conclusions were careful. I agree with them.

References

The references are relatively recent. Up to 5 years (starting in 2019) has approximately 25% and up to 3 years (from 2020) we have approximately 15%. I think it's important to stay around 20% at least. However, many references are from 2017 and 2018. So, I consider recent references.

Author Response

Dear Reviewer,

Thank you very much for your valuable contributions to our paper: we tried incorporating your suggestions to our best. Below, a list of key-points we addressed for resubmission:

  1. We have added on lines 57-59 more zein properties. We also cited another study on articular cartilage scaffolds from e-spun PLGA/nHA/zein. I will also consider using other manufacturing technologies suggested in your comment.  
  2. Table 1 was moved to the Materials and Methods section. The figures with FTIR-ATR and WCAs were separated for better comprehension. 
  3. Regarding the concerns with fiber measurements, the scales reported on top of the images are 10 micrometers, and the intervals are in nanometers.  We believe this visualization on histograms is appropriate and aligned with the electrospinning literature. To reduce interpretation ambiguities, we enlarged the scale and included a note on the figure legend. 
  4. The cell passaging (P1) and other references to the cells were corrected. That was a valuable contribution to eliminate interpretation mistakes. 
  5. No posterior tests were made after UV sterilization. However, this is a valid concern and perhaps a side project. We also included that sterilization was performed in the safety cabinet on l. 157-158. 
  6. Finally, regarding protein adsorption, we only tested protein adsorption as described in the protocol. It is in our knowledge that protein-adsorption is a kinetic adsorption/desorption process. That could also be further explored.
  7. Despite the slightly higher measured fluorescence in the zein-comprised scaffolds, the measurements' differences were not statistically significant. Therefore, we opted for leaving the results in the supplementary information. 

Once again, thank you for the detailed comments and suggestions. They reflect interesting new perspectives and will stimulate future work.

Best Regards,

Reviewer 2 Report

This is a well-done study about the nanofiber surface of poly(ε-caprolactone)-zein-based one-nozzle electrospun articular cartilage scaffolds. I strongly suggest it for publication on Bioengineering after the following minor points are addressed.

1. Line 51-52, one more recent review (10.1002/adma.202005513) should be included to support such a claim. 

2. Line 70, the information about the PDI of PCL should be added.

3. Figure 5, it is better to normalize the protein adsorption in order to compare the results with other studies.

Minor editing of English language required

Author Response

Dear Reviewer,

Thank you very much for your feedback and suggestions on our manuscript. We tried our best to incorporate all the points that were brought up. We added the suggested reference in lines 51-52. For the PDI (Polydispersity index), we obtained the polymers as described by Sigma Aldrich (catalog number 440744) and they do not inform in the PDI values in the Technical Information. Finally, for the normalisation of protein adsorption, it was rather challenging adapting the protocols for finding the range of adsorption in the electrospun materials. We went to several protocol optimization protocols; therefore, we opted to report the data as it is in the text. We are open to further clarification, should it be necessary.

Best Regards, 

Reviewer 3 Report

The manuscript reports a kind of electrospun nanofibers, which was composed of PCL and zein for a better integration with the surrounding soft tissue and fine tribological properties for potential treating of osteoarthritis. A series of characterization methods were explored for evaluation. In general, the topic is interesting and the contents fall within the scope of BIOENGINEERING. I recommend its acceptance for publication after some revisions.

The title can be simple as Electrospun poly(ε-caprolactone)-zein fibrous films as articular cartilage scaffolds”.

electrospun-based is not an appropriate phrase, which can be replaced by nanofiber-based ??????, or electrospun ??????.

The INTRODUCTION can be re-written to provide a timely background of electrospinning, such as coaxial electrospinning and core-sheath nanofibers (10.3390/bioengineering5030068), side-by-side electrospinning and Janus nanofibers (doi:10.1016/j.apmt.2023.101766), tri-fluid electrospinning and tri-layer core-shell or Janus nanofibers (10.3390/bioengineering10010009), production on a large scale and the related energy saving and cutting costs issues (doi:10.3390/polym12102421). The merits of your job can be melt into these electrospinning processes and the resultant nanofibers.

In Figure 1, the scale bars are not clear, please enlarged them.

In Figure 3, the FTIR spectra curve lines please be separated for a clear reding.   

In Figure 4, it is better to use the same font in one Figure.

What if the probing depth in the FTIR-ATR characterization.  

English writing can be improved.

The references format pleae be unified and the most recent years references are too small. To relate your articles contents with the most recent development can benefit a high impact of your article after publication.   

Minor revision is needed.

Author Response

Dear Reviewer,

Thank you for the suggestions and comments made to our manuscript. We tried our best to incorporate all of your feedback. Therefore, we adapted the title of the manuscript. Your suggestions made to Figure 1,3,4 were all taken: the scale bars in Figure 1 were enlarged to double their original size and a reference to the units of measurement was added to the figure legend. The FTIR-ATR spectra were all plotted separately, received new labels, and split from the water contact angles. Addressing your questions: the penetration depth of most commercial FTIR devices is from approx. 0.5-5 micrometers (https://mmrc.caltech.edu/FTIR/Literature/ATR/Intro%20to%20ATR.pdf), which is significantly higher compared to the 10 nm of XPS (https://www.sciencedirect.com/topics/agricultural-and-biological-sciences/x-ray-photoelectron-spectroscopy). As pointed out by another reviewer 25% of references are up to 5 years and 15% up to 3 years, which was considered recent.

Finally, we decided not to expand the introduction to the suggested electrospinning background to focus on the effects of zein on the electrospun surface. Furthermore, electrospinning advantages recent comprehensive reviews have been published elsewhere. The suggested topics (e.g., Janus nanofibers) were already discussed by existing literature in other review papers (doi:10.1016/j.apmt.2023.101766), for example. We decided to focus on the surface segregation of zein incorporation in the discussion (lines 449-450, 458-459). A good review of the topic was referenced there (do: 10.1080/09506608.2018.1484577).

Best Regards,